# Knowledge, attitudes and practices regarding bovine tuberculosis in cattle and humans in Malawi

**Alfred Ngwira**[1,2,3]*, **Samuel O.M. Manda**[4], **Esron D. Karimuribo**[1,3], **Sharadhuli I. Kimera**[1,3]

**1** Department of Veterinary Medicine and Public Health, Sokoine University of Agriculture, Morogoro, Tanzania, **2** Department of Basic Sciences, Lilongwe University of Agriculture and Natural Resources, Lilongwe, Malawi, **3** SACIDS Foundation for One Health, SACIDS Africa Centre of Excellence for Infectious Diseases, Sokoine University of Agriculture, Morogoro, Tanzania, **4** Department of Statistics, University of Pretoria, Pretoria, South Africa

* alfred.ngwira@sacids.org

## Abstract

### Background

Bovine tuberculosis (BTB) has a significant impact on both the cattle industry and human health. Understanding its transmission, clinical signs, risk factors, and control measures is essential for prevention. This study aimed to evaluate the knowledge, attitudes, and practices regarding BTB in high-burden areas of Malawi.

### Methods

A stratified sampling design was employed to collect data from 463 butchers, dairy farmers and cattle farmers in different locations and settings in Malawi. Aggregate scores on knowledge, attitudes, and practices were taken as multivariate measurements and fitted with multivariate linear regression models.

### Results

Participants displayed satisfactory knowledge (88.68%), negative attitudes towards risky behaviours (92.27%), and appropriate practices (86.83%) concerning BTB. Most were aware of its presence in cattle (85.71%) and potential transmission to humans (74.09%), yet exhibited gaps in understanding clinical signs and held misconceptions about hereditary transmission. Many participants demonstrated risky behaviours, such as consuming raw animal products and selling products from infected animals. Individuals with secondary (β: −2.148; 95% CI: −4.168, −0.127) and tertiary education (β: −3.488; 95% CI: −6.626, −0.349) exhibited more negative attitudes towards risky behaviours compared to those with informal education. Adults aged 18−30 (β: −2.777; 95% CI: −5.469, −0.085) and those aged 31−45 (β: −3.035; 95% CI: −5.752, −0.319) demonstrated better protective practices than youths under 18.

**Data availability statement:** The data are available from figshare at https://doi.org/10.6084/m9.figshare.29815925.v1.

**Funding:** This study was supported by the Regional Scholarship and Innovation Fund (RSIF) of the Partnership for Skills in Applied Sciences, Engineering and Technology (PASET) (Project Grant No. P165581) grant to SACIDS Africa Centre of Excellence for Infectious Diseases of Humans and Animals in Southern and East Africa (SACIDS-ACE) at the Sokoine University of Agriculture (SUA). Alfred Ngwira was a recipient of an RSIF-PASET doctoral scholarship at SUA.

**Competing interests:** The authors have declared that no competing interests exist.

## Conclusion

There is an urgent need for targeted health education on BTB, focusing on clinical signs and the myth of hereditary transmission, particularly aimed at young people, individuals with limited formal education, and farmers, to enhance prevention efforts.

---

## Introduction

Bovine tuberculosis (BTB) is a significant infectious disease primarily affecting cattle and other bovine species, with *Mycobacterium bovis (M. bovis)* as the primary causative agent. In Malawi, the prevalence of bovine tuberculosis in the cattle population is notably high at 9.95% [1], emphasising the greater challenges faced within the sub-Saharan African region [2]. While the impact of BTB on human health is not well documented [3], a focused study in a specific area estimates a prevalence of 3.3% [4], indicating a concerning burden that warrants attention.

Bovine tuberculosis poses a serious economic threat to the cattle industry, leading to significant financial losses from meat condemnations in abattoirs [5]. Furthermore, in humans, the diagnosis and treatment of BTB are complicated by the disease's drug resistance [6] and the often low bacterial load detected in clinical samples [7,8].

The transmission of *M. bovis* occurs through various bodily fluids, including exhaled air, nasal discharge, milk, urine, and reproductive secretions [9]. In cattle, the primary mode of transmission is through close contact in environments such as intensive farming setups, communal waterholes, or bustling marketplaces. For humans, infection can occur through the consumption of raw or undercooked animal products, inhalation of aerosols containing the bacteria, or direct contact with contaminated animal skin. In cattle, clinical symptoms manifest as weakness, persistent low-grade fever, loss of appetite, emaciation, rough and unkempt fur, a chronic cough, swollen lymph nodes, laboured breathing, and tachypnoea [10]. Similarly, infected humans may experience a range of alarming symptoms, including night sweats and chest pain [10].

Several risk factors increase cattle's susceptibility to BTB, including being of a foreign breed, older age, intensive farming practices, herd movements, trading activities, the introduction of new cattle into a group, and interactions with wildlife [10]. For humans, the risk increases with factors such as low immunity, age-related vulnerabilities, consumption of raw animal products, employment in slaughterhouses, and exposure to non-bovine species such as goats [10].

Evaluating knowledge, attitudes, and practices (KAP) related to bovine tuberculosis is an essential component of efforts to prevent infections in both cattle and humans. Insights from KAP assessments can inform public health officials about the need for targeted health campaigns to reduce the spread of BTB. According to KAP theory [11], enhancing knowledge fosters favourable attitudes towards disease prevention, which, in turn, can lead to positive behaviours. In this context, individuals lacking awareness of BTB—including its transmission routes, clinical features, and associated risk factors—are likely to develop positive attitudes towards risky behaviours, ultimately leading to practices that increase the risk of infection.

The assessment of knowledge regarding bovine tuberculosis (BTB) in northern Malawi revealed a significant level of disease awareness at 74.3% [12]. However, information concerning high-burden areas in the central and southern regions remains limited. Our study aimed to evaluate the knowledge, attitudes, and practices (KAP) related to BTB, focusing on both animal and human infection aspects. Guided by the KAP theory [11] and prior research findings [12], we postulate two hypotheses: a) a high level of knowledge, attitudes, and practices is anticipated, and b) a negative correlation exists between knowledge and attitude, as well as between knowledge and practices. Given the presumed interrelation among the KAP elements, we will employ multivariate regression analysis rather than conducting separate outcome regressions to identify the determinants of KAP. The advantages of employing multivariate analysis include reduced standard errors and the ability to compare covariate effects across multiple outcomes [13].

## Materials and methods

### Study area and population

The study was conducted in purposefully selected areas in Malawi's central and southern regions (S1 Fig). We included the selected areas due to the fact that they were known to have higher burden of BTB [3,14]. In the central region, we targeted dairy farmers in Lilongwe rural and butchers in Lilongwe city. Dairy farmers were from Nanthenje, Machite, Lumbadzi, and Mkwinda milk bulking groups (S1 Fig), while butchers were from abattoirs and butchery shops. In the southern region, the study population included cattle farmers near Majete Wildlife Reserve in Chikwawa District, specifically at Maganga, Kaisi, and Bwalo (S1 Fig). The source of the shapefiles for the map of study areas (S1 Fig) was NSO/Malawi and OCHA Field Information Services Section [15].

### Sample size, sampling and data collection

The study design was cross-sectional, with a purposive stratified sampling technique, where the strata were butchers, dairy farmers, and farmers near a wildlife reserve. Since the study aimed to estimate the proportion of participants with appropriate BTB knowledge, based on the prevalence estimated from a previous study [12] and the sample size calculation formula for prevalence studies [16], the estimated minimum sample size was 463. Due to the absence of total population for each stratum, we allocated the samples equally to the strata, resulting in 154 participants per stratum.

Dairy farmers were selected using convenience cluster sampling, with the clusters being the milk bulking groups, selected from the list provided by the Central Region Milk Producers Association (CREMPA). The selection of milk bulking groups depended on the chairperson's consent. All eligible dairy farmers in the selected milk bulking groups were studied at their homes. Eligible dairy farmers were those who were milking at the time of the survey and those who were not milking, but had milked before. Selection of butchers was by convenience and purposive sampling, where those that were willing and selling beef were selected, and were studied either at abattoirs or at their butchery shops. Cattle farmers in the southern region were also purposively sampled, with farmers who owned cattle and those close to the game reserve included. Cattle farmers in the southern region were studied at their homes.

Data collection was via a questionnaire survey with items assessing participants' knowledge, attitudes, and practices (S1 Appendix). The questionnaire design was based on similar previous studies [12,17,18]. Knowledge items were aimed to assess participants' awareness of animal TB in cattle, including its zoonotic potential, clinical signs, risk factors, and prevention. Questions on attitudes were intended to assess participants' perceptions, such as those on disease fatality. The assessment of participants' practices included their engagement in risky activities, such as the consumption of raw animal products. Item responses on knowledge and practices were coded as 1 (yes), 2 (no), and 3 (don't know), while those on attitudes were coded as 1 (strongly disagree), 2 (disagree), 3 (neutral), 4 (agree), and 5 (strongly agree). A "yes" response on knowledge items corresponded to correct knowledge, and a "no" translated to wrong knowledge. A "yes" on practice items indicated bad practice, and a "no" corresponded to good practice. A lower value on the Likert scale for

attitude items indicated a good or negative attitude, and a higher value corresponded to a bad or positive attitude. Data was also collected on predictor variables such as age, education, and income [12,17,18]. Participants' income was measured through their responses to the amount of money they could approximately make from different economic activities in a month. The questionnaire was administered in person, with a soap gift offered after the interview. A questionnaire in the local language was used during data collection (S2 Appendix). Ethical clearance to conduct the study in Malawi was granted by the Department of Animal Health and Livestock Development (DAHLD) (#DAHLD/AHC/10/2022/1) (S3 Appendix). The DAHLD issues ethical clearance for diseases that fall between humans and animals, such as bovine tuberculosis. The verbal consent-to-participate process was documented with a "yes" or "no" tick mark in the questionnaire's introduction section (S1 Appendix) and involved no witnesses. The DAHLD approved the verbal consent process.

## Statistical analysis

We reassigned a zero to "no" and "don't know" responses on knowledge and practice items [19]. The overall score for each participant on each sub-theme was calculated by summing the item responses [20]. Based on the measurement scales for knowledge, attitude, and practice items, a high total score on knowledge indicated good knowledge, while a high total score on attitude and practice indicated a poor attitude or poor practices. The knowledge scores were further classified as poor (<50%), good (50% to < 75%), and excellent (≥75%), and the classes of attitude and practice scores were poor (>50%), good (>25% to ≤ 50%), and excellent (≤25%) [19]. Participants in the "poor" category were considered to have inappropriate knowledge, attitudes, or practices, and appropriate otherwise. We tabulated the categorised knowledge, attitude, and practice scores, including individual question items, to have their frequency distributions.

A multivariate linear regression of the scores was fitted to identify factors related to participants' knowledge, attitudes, and practices regarding bovine tuberculosis. Denoting the score for participant $i = 1, 2, 3, \ldots, n$, as $y_{ik}$, where $k = 1, 2,$ and $3$ were knowledge, attitude, and practice, respectively, the multivariate linear regression was defined as $y_{ik} = x'_{ik}\beta_k + e_{ik}$, where $x$ represented demographic characteristics, and $e_{ik}$ were the residuals. The residuals were assumed to have the multivariate normal distribution $MVN(0, \Omega_e)$, where $\Omega_e$ was the variance-covariance matrix with variances and pairwise correlations. The multivariate model was fitted so as to account for correlations among knowledge, attitudes, and practices and to improve estimation efficiency. A t-test was used to compare covariate coefficients between outcomes, and standard errors of coefficients between univariate and multivariate regression to assess efficiency. Data analysis was performed using the multivariate regression (*mvreg*) routine in Stata.

## Results

The mean score of knowledge was 18.58±4.96, with a minimum of zero and a maximum of 26 out of 26. Most participants demonstrated appropriate knowledge (88.68%), with 37.74% and 50.94% having good and excellent knowledge, respectively (S2 Fig). The distribution of specific knowledge items (S1 Table) indicates most participants were aware that BTB infects cattle (85.71%), including its transmission to humans (74.09%). Participants were also aware of most of the risk factors in cattle, such as having a kraal with poor ventilation, intensive farming, and animal-wildlife interactions at waterholes. Participants were also aware of the majority of risk factors for humans, including consumption of raw animal products, meat handling, and dairy farming. Knowledge of control and prevention measures such as test and slaughter (83.41%) and use of education campaigns (93.88%) was also good. Regardless of good overall BTB knowledge, participants had limited knowledge on whether being in contact with infected livestock, risks human infection (56.69%). A limited proportion of participants (37.87%) answered "yes," correctly recognizing that BTB is not transmitted through heredity. Participants also demonstrated limited knowledge on clinical signs, such as lymph node enlargement, weight loss, and low-grade fever.

The average of attitude scores was 13.82±4.60, with a minimum of 8 and a maximum of 33 out of 40. Many participants demonstrated an appropriate attitude (92.27%), representing 65.68% and 26.59% with good and excellent attitudes,

respectively (S2 Fig). Specifically (S2 Table), most participants had negative perceptions towards drinking or eating and selling products of an infected animal, including the perception that BTB is fatal. Nevertheless, a considerable proportion of participants had a positive perception of living together with sick animals or people, including not being afraid of them. A substantial proportion of participants also had the perception that people with BTB have human immunodeficiency virus (HIV).

The average practice score was 4.27 ± 2.05, with a minimum of zero and a maximum of 11 out of 12. Overall, a high proportion of participants demonstrated appropriate practices (86.83%), where 49.88% and 36.95% showed good and excellent practice behaviour, respectively (S2 Fig). Specific good practices (S3 Table) involved not drinking or eating raw animal products, and not selling animal products from a sick animal. Despite overall good and excellent participants' behaviour, a notable proportion of participants were not using protective wear when handling animal products, including mixing cattle from different households during grazing, grazing animals close to a wildlife reserve, keeping animals in the house, and drinking or eating raw meat and milk. Furthermore, a substantial proportion of participants engaged in treatment malpractices, such as arriving late at the hospital, buying medicine from the pharmacy before going to the hospital, and buying medicine for livestock before calling the veterinary officer.

Tables 1 and 2 show the results of univariate and multivariate regression of knowledge, attitude, and practice scores on demographic characteristics. There was no significant difference in the model efficiency between univariate and multivariate regression (S4 Table). Participants with secondary and higher education had lower attitude scores than those without formal education, suggesting that secondary and higher education were associated with more negative perceptions of engaging in risky behaviours than non-formal education. The effect of secondary and higher education was different between attitudes and practices (Table 2, S4 Table), with education having a greater negative impact on perceptions than on behaviours. Increased age (18 and above) and being a student were negatively associated with BTB practice score. This means that adults aged 18 and above had better practices than youths aged less than 18, and students were associated with good practices compared to farmers. The variance-covariance matrix of the residuals (S5 Table) indicated a significant negative correlation between knowledge and attitude (*r*: −0.139; *p*: 0.004), suggesting that greater knowledge about BTB was associated with more negative feelings towards engaging in risky behaviours. There was a positive correlation between attitude and practice (*r*: 0.218; *p*: < 0.001), translating to positive feelings towards engaging in risky behaviours, encouraged malpractices.

## Discussion

This study investigated the level of knowledge, attitudes, and practices regarding BTB infection in cattle and humans, with a focus on high-burden areas in Malawi. We used data from a stratified sample survey involving butchers and dairy farmers from the central region, including cattle farmers from the southern region. Overall, participants demonstrated good knowledge, attitudes, and practices regarding BTB. Nevertheless, a notable proportion of participants demonstrated poor knowledge of certain clinical signs, such as lymph node enlargement, held positive perceptions of living with sick animals or people, and engaged in malpractice by failing to wear protective equipment, sharing a house with livestock, and consuming raw animal products. Participants with secondary and higher education were associated with a more negative attitude towards engaging in risky behaviours than those without formal education. Adults aged 18 and above, and students, demonstrated better practices than youths and farmers, respectively.

A high level of overall knowledge about BTB among participants may be attributed to the educational information they receive from health workers, veterinary officers, and the mass media [21,22], including peer interactions in groupings such as milk bulking groups. The high level of awareness of BTB infection among participants is consistent with the literature [12]. Increased awareness that BTB is transmitted through the air is consistent with a human tuberculosis study in Malawi [22], in which most participants reported that tuberculosis is spread through the air when coughing or sneezing. Poor knowledge of certain clinical signs may be due to limited formal education among

**Table 1.** Univariate linear regression of knowledge, attitudes and practices about BTB.

| Independent variable | Knowledge | | Attitude | | Practices | |
|---|---|---|---|---|---|---|
| | Coefficient (95% CI) | se | Coefficient (95% CI) | se | Coefficient (95% CI) | se |
| Age | | | | | | |
| <18 (Ref.) | – | – | – | – | – | – |
| 18–30 | 3.466 (−3.210, 10.143) | 3.396 | 1.345 (−4.780, 7.469) | 3.116 | −2.710* (−5.398, −0.022) | 1.367 |
| 31–45 | 4.361 (−2.368, 11.091) | 3.423 | −0.113 (−6.288, 6.063) | 3.141 | −3.017* (−5.730, −0.303) | 1.380 |
| >45 | 4.691 (−2.026, 11.408) | 3.417 | −0.717 (−6.881, 5.448) | 3.136 | −2.546 (−5.254, 0.162) | 1.377 |
| Sex | | | | | | |
| Male (Ref.) | – | – | – | – | – | – |
| Female | −0.449 (−1.571, 0.672) | 0.571 | −0.315 (−1.329, 0.699) | 0.516 | −0.272 (−0.719, 0.175) | 0.227 |
| Education | | | | | | |
| No education (Ref.) | – | – | – | – | – | – |
| Primary | 0.342 (−1.644, 2.328) | 1.010 | −0.655 (−2.43, 1.121) | 0.903 | −0.349 (−1.138, 0.439) | 0.401 |
| Secondary | 1.424 (−0.758, 3.606) | 1.109 | −2.165* (−4.116, −0.213) | 0.993 | −0.046 (−0.912, 0.821) | 0.441 |
| Higher | 0.818 (−2.499, 4.135) | 1.687 | −2.985* (−5.951, −0.019) | 1.509 | 0.108 (−1.236, 1.452) | 0.684 |
| Occupation | | | | | | |
| Farmer (Ref.) | – | – | – | – | – | – |
| Student | 0.591 (−5.769, 6.951) | 3.235 | −2.145 (−7.941, 3.651) | 2.949 | −3.461* (−6.006, −0.915) | 1.295 |
| Business | 1.919 (−1.776, 5.613) | 1.879 | −0.831 (−4.028, 2.365) | 1.626 | −0.520 (−1.924, 0.884) | 0.714 |
| Employed | 2.583 (−1.124, 6.291) | 1.886 | −1.246 (−4.471, 1.978) | 1.640 | −0.110 (−1.527, 1.307) | 0.721 |
| Labourer | 0.161 (−3.929, 4.252) | 2.081 | 1.480 (−2.121, 5.082) | 1.832 | 0.012 (−1.570, 1.594) | 0.805 |
| Residence | | | | | | |
| Urban (Ref.) | – | – | – | – | – | – |
| Rural | 0.263 (−2.282, 2.808) | 1.295 | 0.961 (−1.305, 3.227) | 1.153 | 0.853 (−0.142, 1.849) | 0.506 |
| Marriage | | | | | | |
| Yes (Ref.) | – | – | – | – | – | – |
| No | 1.370 (−0.048, 2.789) | 0.722 | −0.588 (−1.848, 0.672) | 0.641 | −0.138 (−0.693, 0.417) | 0.282 |
| Income (MK) | | | | | | |
| <100 (Ref.) | – | – | – | – | – | – |
| 100–500 | 0.784 (−0.593, 2.160) | 0.700 | −0.108 (−1.356, 1.139) | 0.635 | −0.024 (−0.574, 0.527) | 0.280 |
| >500 | 0.806 (−1.961, 3.573) | 1.407 | 1.540 (−0.917, 3.997) | 1.249 | 1.067 (−0.032, 2.165) | 0.559 |
| Risk group | | | | | | |
| Butchers (Ref.) | – | – | – | – | – | – |
| Dairy farmers | 2.904 (−1.129, 6.938) | 2.052 | −1.174 (−4.737, 2.388) | 1.812 | −0.041 (−1.605, 1.523) | 0.796 |
| Farmers near wildlife | 2.177 (−1.917, 6.271) | 2.083 | −1.507 (−5.120, 2.107) | 1.838 | −0.719 (−2.307, 0.868) | 0.807 |

\* Significant at 5% significance level; CI: Confidence interval; Ref.: Reference; se: Standard error; MK: Malawi kwacha.

participants, as education is usually positively associated with BTB knowledge [23,24]. Limited awareness of BTB clinical signs has also been documented in the literature, especially among dairy farmers [25–27]. Similar limited knowledge of clinical signs, such as lymph node enlargement, has been documented in a KAP study about human tuberculosis in Malawi [28]. Furthermore, increased awareness of coughing as a BTB clinical sign is consistent with its recognition as a clinical sign of human TB in Malawi [29], where most participants mentioned coughing as a sign of human tuberculosis.

**Table 2. Multivariate linear regression of knowledge, attitudes and practices about BTB.**

| Independent variable | Knowledge | | Attitude | | Practices | |
|---|---|---|---|---|---|---|
| | Coefficient (95% CI) | se | Coefficient (95% CI) | se | Coefficient (95% CI) | se |
| Age | | | | | | |
| <18 (Ref.) | – | – | – | – | – | – |
| 18–30 | 3.444 (−3.291, 10.179) | 3.426 | 1.441 (−4.670, 7.552) | 3.108 | −2.777* (−5.469, −0.085) | 1.369 |
| 31–45 | 4.419 (−2.379, 11.216) | 3.457 | 0.049 (−6.118, 6.217) | 3.137 | −3.035* (−5.752, −0.319) | 1.382 |
| >45 | 4.779 (−2.003, 11.563) | 3.450 | −0.371 (−6.526, 5.783) | 3.131 | −2.531 (−5.242, 0.180) | 1.379 |
| Sex | | | | | | |
| Male (Ref.) | – | – | – | – | – | – |
| Female | −0.467 (−1.603, 0.669) | 0.578 | −0.264 (−1.295, 0.767) | 0.524 | −0.281 (−0.735, 0.173) | 0.231 |
| Education | | | | | | |
| No education (Ref.) | – | – | – | – | – | – |
| Primary | 0.409 (−1.619, 2.438) | 1.032 | −0.557 (−2.398, 1.283) | 0.936 | −0.453 (−1.264, 0.357) | 0.412 |
| Secondary | 1.547 (−0.679, 3.774) | 1.132 | −2.148* (−4.168, −0.127) | 1.028 | −0.082 (−0.972, 0.808) | 0.453 |
| Higher | 0.882 (−2.577, 4.340) | 1.759 | −3.488* (−6.626, −0.349) | 1.596 | 0.048 (−1.334, 1.431) | 0.703 |
| Occupation | | | | | | |
| Farmer (Ref.) | – | – | – | – | – | – |
| Student | 0.639 (−5.783, 7.061) | 3.266 | −1.774 (−7.60, 4.053) | 2.964 | −3.216* (−5.783, −0.649) | 1.305 |
| Business | 1.875 (−1.852, 5.603) | 1.896 | −0.343 (−3.725, 3.039) | 1.720 | −0.039 (−1.529, 1.451) | 0.758 |
| Employed | 2.615 (−1.128, 6.357) | 1.903 | −0.922 (−4.317, 2.474) | 1.727 | 0.361 (−1.135, 1.857) | 0.761 |
| Labourer | 0.189 (−3.938, 4.318) | 2.099 | 2.081 (−1.665, 5.827) | 1.905 | 0.466 (−1.184, 2.116) | 0.839 |
| Residence | | | | | | |
| Urban (Ref.) | – | – | – | – | – | – |
| Rural | 0.298 (−2.273, 2.869) | 1.308 | 0.319 (−2.013, 2.652) | 1.187 | 0.943 (−0.085, 1.970) | 0.523 |
| Marriage | | | | | | |
| Yes (Ref.) | – | – | – | – | – | – |
| No | 1.372 (−0.064, 2.808) | 0.730 | −0.378 (−1.681, 0.924) | 0.663 | −0.097 (−0.671, 0.477) | 0.292 |
| Income (MK) | | | | | | |
| <100 (Ref.) | – | – | – | – | – | – |
| 100–500 | 0.728 (−0.669, 2.125) | 0.710 | 0.056 (−1.212, 1.323) | 0.645 | 0.000 (−0.558, 0.558) | 0.284 |
| >500 | 0.592 (−2.254, 3.439) | 1.448 | 1.665 (−0.918, 4.248) | 1.314 | 0.809 (−0.328, 1.947) | 0.579 |
| Risk group | | | | | | |
| Butchers (Ref.) | – | – | – | – | – | – |
| Dairy farmers | 2.847 (−1.223, 6.916) | 2.069 | −0.199 (−3.892, 3.494) | 1.878 | 0.425 (−1.201, 2.052) | 0.827 |
| Farmers near wildlife | 2.188 (−1.944, 6.319) | 2.102 | −0.557 (−4.306, 3.193) | 1.907 | −0.305 (−1.957, 1.346) | 0.840 |

* Significant at 5% significance level; CI: Confidence interval; Ref.: Reference; se: Standard error; MK: Malawi kwacha.

A negative attitude towards engaging in risky behaviours, such as not eating or selling infected meat, could stem from participants' high knowledge of BTB risk factors, which was driving their negative attitude towards risky behaviours [11], as evidenced by the observed negative correlation between knowledge and attitude. The substantial proportion of participants demonstrating the perception that BTB is fatal and that humans who have BTB, have HIV, is in agreement with a human tuberculosis study [29], where participants viewed TB as dangerous before diagnosis, and that those who are infected with TB look thin, similar to those with HIV.

The practice of keeping animals in the house was reported to be attributed to security concerns, especially among farmers with small ruminants, such as goats and sheep. In this regard, even though BTB mainly infect cattle, goats may also be infected [30], and henceforth poses a risk of infection in humans. A previous study found that sharing premises with livestock, including not using protective gear when handling animal products, was common [18]. Drinking of raw milk could be influenced by food traditions and low education levels [26], while eating raw or undercooked meat might be the issue of gender, where more males might be involved [31]. A considerable proportion of participants going to the hospital late could be related to their fear of testing positive for HIV, and their tendency to initially consult witch doctors before seeking professional health care [32].

A good attitude among participants with secondary and higher education could be attributed to improved knowledge of disease facts through formal education, which, in turn, improved their perceptions of risky behaviours, according to the KAP theory [11]. Better BTB practices among adults aged 18 and above have been consistent with the literature [33], where adults aged 30–59 were purported to have better practices than the younger because of their easier access to information on the internet. Better practices among adults than among youths could also be attributed to long-standing practical knowledge and experience [24]. Good practices among students rather than farmers may be attributed to students' higher levels of formal education, which negatively affect their perceptions of risk and behaviours. The observed significant correlations between KAP sub-themes are consistent with KAP theory [11], which posits that knowledge influences attitudes, which, in turn, influence behaviours, with no significant relationship between knowledge and practices.

Weaknesses of the study included not establishing a control group with a low BTB burden, thereby preventing comparison of knowledge, attitudes, and practices between the control and high-burden areas. In addition, the observed correlations did not imply causal relationships, since this was a cross-sectional study. Despite these shortcomings, the study's findings are consistent with a previous study [12] and the KAP theory [11].

## Conclusion

Participants had limited knowledge of clinical signs, including the myth of heredity as the mode of transmission. A considerable proportion of participants had the perception that if humans have bovine tuberculosis, they have human immunodeficiency virus, including the belief of living together and not fearing people or animals with bovine tuberculosis. A substantial proportion of participants engaged in malpractice, including eating or drinking raw animal products and sharing a house with animals. Secondary and higher education were associated with negative perceptions towards engaging in risky behaviours. Adults aged 18 and above and students had better practices than youths aged less than 18 and farmers, respectively. Health education about bovine tuberculosis should focus on clinical signs and the myth of heredity as a mode of transmission, while targeting the youths, informally educated, and farmers.

## Supporting information

**S1 Fig. Study areas of knowledge, attitudes and practices about BTB (Source of shapefiles: NSO/Malawi and OCHA Field Information Services Section (https://data.humdata.org/dataset/cod-ab-mwi)).**
(DOCX)

**S2 Fig. Distribution of knowledge, attitudes and practices about BTB.**
(DOCX)

**S1 Table. Percentage of knowledge about BTB in cattle and humans.**
(DOCX)

**S2 Table. Attitudes about BTB in cattle and humans.**
(DOCX)

**S3 Table. Practices about BTB in cattle and humans.**
(DOCX)

**S4 Table. Comparison of standard errors (SE) and coefficients.**
(DOCX)

**S5 Table. Residual correlation matrix and *p*-values.**
(DOCX)

**S1 Appendix. English questionnaire on knowledge, attitudes and practices about BTB in cattle and humans in Malawi.**
(DOCX)

**S2 Appendix. Chichewa questionnaire on knowledge, attitudes and practices about BTB in cattle and humans in Malawi.**
(DOCX)

**S3 Appendix. Approval to collect data on knowledge, attitudes and practices about BTB.**
(DOCX)

## Author contributions

**Conceptualization:** Alfred Ngwira.

**Formal analysis:** Alfred Ngwira.

**Supervision:** Samuel O.M. Manda, Esron D. Karimuribo, Sharadhuli I. Kimera.

**Writing – original draft:** Alfred Ngwira.

**Writing – review & editing:** Alfred Ngwira, Samuel O.M. Manda, Esron D. Karimuribo, Sharadhuli I. Kimera.

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
