## [Decision Letter · Decision Letter 0]

19 Jun 2025

Dear Dr. Ngwira,

We look forward to receiving your revised manuscript.

Kind regards,

Rebecca Lee Smith, D.V.M., M.S., Ph.D.

Academic Editor

PLOS ONE

Journal Requirements:

2. In the ethics statement in the Methods, you have specified that verbal consent was obtained. Please provide additional details regarding how this consent was documented and witnessed, and state whether this was approved by the IRB.

“This study was supported by the Regional Scholarship and Innovation Fund (RSIF) of the Partnership for Skills in Applied Sciences, Engineering and Technology (PASET) (Project Grant No. P165581) grant to SACIDS Africa Centre of Excellence for Infectious Diseases of Humans and Animals in Southern and East Africa (SACIDS-ACE) at the Sokoine University of Agriculture (SUA). Alfred Ngwira was a recipient of an RSIF-PASET doctoral scholarship at SUA. Authors”

5. In this instance it seems there may be acceptable restrictions in place that prevent the public sharing of your minimal data. However, in line with our goal of ensuring long-term data availability to all interested researchers, PLOS’ Data Policy states that authors cannot be the sole named individuals responsible for ensuring data access (http://journals.plos.org/plosone/s/data-availability#loc-acceptable-data-sharing-methods).

7. We note that Supporting Figure S1 in your submission contain map images which may be copyrighted. All PLOS content is published under the Creative Commons Attribution License (CC BY 4.0), which means that the manuscript, images, and Supporting Information files will be freely available online, and any third party is permitted to access, download, copy, distribute, and use these materials in any way, even commercially, with proper attribution. For these reasons, we cannot publish previously copyrighted maps or satellite images created using proprietary data, such as Google software (Google Maps, Street View, and Earth). For more information, see our copyright guidelines: http://journals.plos.org/plosone/s/licenses-and-copyright.

1) You may seek permission from the original copyright holder of Supporting Figure S1 to publish the content specifically under the CC BY 4.0 license.

**Additional Editor Comments:**

Please address the serious concerns from the reviewers around the purpose of a KAP study.

Reviewers' comments:

Reviewer's Responses to Questions

**Comments to the Author**

1. Is the manuscript technically sound, and do the data support the conclusions?

Reviewer #1: No

Reviewer #2: Yes

2. Has the statistical analysis been performed appropriately and rigorously?

Reviewer #1: No

Reviewer #2: I Don't Know

3. Have the authors made all data underlying the findings in their manuscript fully available?

Reviewer #1: Yes

Reviewer #2: Yes

4. Is the manuscript presented in an intelligible fashion and written in standard English?

Reviewer #1: No

Reviewer #2: Yes

Reviewer #1: A KAP survey should assess knowledge, attitudes, and practices directly related to bovine TB. The included variables such as age, sex, education, occupation, marital status, and income are purely demographic and socioeconomic, lacking disease-specific insights. Without assessing awareness, risk perception, and preventive behaviors, the study fails to provide actionable findings for bovine TB control and intervention.

Reviewer #2: - This paper would add critical information regarding farmers and cattle handlers’ knowledge and practices with respect to bovine tuberculosis in Malawi. As it stands, the paper lacks information regarding the various findings obtained from the survey instrument, the survey development and recruitment processes used, discussion of the findings in relation to similar studies in literature, and the implications of this research. I believe addition of these important pieces of information will strengthen the manuscript. I have highlighted areas where additions should be made below.

- Consider adding more information in the abstract: for instance, information regarding the questionnaire used in the methods, along with other important findings obtained in the results sub-section.

- In the abstract, in the sentence ‘Overall knowledge, attitude, and practices about BTB among participants were good’ – how do you define ‘good’?

- In the keywords section, consider adding the keywords ‘KAP survey’ or ‘knowledge, attitude, practice study’.

- The introduction section could do better with additional information and more context on bovine TB. What is the burden of BTB in Malawi as compared to its neighboring countries? How does bovine TB differ from zoonotic TB? Are symptoms similar between bovine and human TB? How does bovine TB occur? What is your hypothesis regarding this particular study population on bovine TB?

- The questionnaire used for this study- was it developed for this study, or has it been adapted from other studies? If so, please provide citations. Were the surveys conducted in person or via mail? Please provide more information on the recruitment and study advertisement processes used. Did you provide any incentives to participants as part of the study? What was the language in which the KAP survey was administered in? Were translation and back-translation processes needed for the survey?

- Which software/platform did you use to perform the statistical analyses?

- I am curious, is there any reason as to why you chose to show that higher scores for knowledge corresponds with more knowledge whereas higher scores in attitudes and practices corresponds to low perceptions and practices?

- I think it would be helpful and informative if you added separate findings obtained for the knowledge, attitude, and practice sections of the survey instrument in the results section, in addition to what is currently presented.

- How do the results from this KAP study stand in relation to other KAP studies on human/bovine TB? Are these results similar or different?

- Please discuss in more detail the interpretation of the results and regression analyses and the implications of these findings in context with other studies on this topic. Was the KAP survey successful in capturing knowledge gaps and data gaps among the surveyed participants in Malawi?

- What would be some future directions as an outcome of this study?

**Do you want your identity to be public for this peer review?** For information about this choice, including consent withdrawal, please see our Privacy Policy

Reviewer #1: No

Reviewer #2: No

---

## [Author Response · Author response to Decision Letter 1]

20 Aug 2025

RESPONSE TO REVIEWERS

Reviewer #1: A KAP survey should assess knowledge, attitudes, and practices directly related to bovine TB. The included variables such as age, sex, education, occupation, marital status, and income are purely demographic and socioeconomic, lacking disease-specific insights. Without assessing awareness, risk perception, and preventive behaviours, the study fails to provide actionable findings for bovine TB control and intervention.

Response: The disease specific questions are the in supplementary file in the questionnaire (S1 Appendix) and their results are in supplementary tables S1-S3 Tables. We suspect the reviewer did not check the supplementary information. The mentioned demographic characteristics were used to determine factors of KAP. This is a common practice when collecting KAP data where disease specific questions plus demographic characteristics are collected with the aim of determining their influence on KAP. See e.g Hamid et al (2024) www.onehealthjournal.org/Vol.10/No.1/11.pdf. Nevertheless, the paper has been updated in the “Abstract”, “’Results” and “discussion”’ sections detailing findings on disease specific questions.

Reviewer #2: - This paper would add critical information regarding farmers and cattle handlers’ knowledge and practices with respect to bovine tuberculosis in Malawi. As it stands, the paper lacks information regarding the various findings obtained from the survey instrument, the survey development and recruitment processes used, discussion of the findings in relation to similar studies in literature, and the implications of this research. I believe addition of these important pieces of information will strengthen the manuscript. I have highlighted areas where additions should be made below.

- Consider adding more information in the abstract: for instance, information regarding the questionnaire used in the methods, along with other important findings obtained in the results sub-section.

Response: We have added the extra result information in the abstract. Page 1, line 29-34.

- In the abstract, in the sentence ‘Overall knowledge, attitude, and practices about BTB among participants were good’ – how do you define ‘good’?

Response: We added the percentage statistics to show how good it was. Page 1, line 29-34.

- In the keywords section, consider adding the keywords ‘KAP survey’ or ‘knowledge, attitude, practice study’.

Response: We have added the key word “KAP survey”. Page 2, line 2,3.

- The introduction section could do better with additional information and more context on bovine TB. What is the burden of BTB in Malawi as compared to its neighboring countries? How does bovine TB differ from zoonotic TB? Are symptoms similar between bovine and human TB? How does bovine TB occur? What is your hypothesis regarding this particular study population on bovine TB?

Response: We have added information about BTB burden in Malawi and how is related to neighbours. Page 2, line 7-9.

- The questionnaire used for this study- was it developed for this study, or has it been adapted from other studies? If so, please provide citations. Were the surveys conducted in person or via mail? Please provide more information on the recruitment and study advertisement processes used. Did you provide any incentives to participants as part of the study? What was the language in which the KAP survey was administered in? Were translation and back-translation processes needed for the survey?

- Which software/platform did you use to perform the statistical analyses?

Response: We have added all the requested information about the survey methodology on page 3-4. i.e

1) Citation of where we adapted question items for the questionnaire is indicated on Page 3, line 35, 36.

2) Whether survey was in person/email is indicated on Page 4, line 4, 5.

3) Provision of incentives is indicated on Page 4, line 5.

4) Language used in the questionnaire is indicated on page 4, line 5,6.

5) Software used is indicated on page 4, line 32,33.

- I am curious, is there any reason as to why you chose to show that higher scores for knowledge corresponds with more knowledge whereas higher scores in attitudes and practices corresponds to low perceptions and practices?

Response: It was not by choice, but by the measurement scales of specific items for knowledge (0,1), attitude (1,2,3,4,5) and practices (0,1). In this regard, a one for knowledge meant correct knowledge and a zero corresponded to wrong or don’t know. A lower value on the attitude Likert scale (e.g 1,2) corresponded to negative or good attitude about risky behaviours, and a higher value e.g 4, 5, indicated positive or bad attitude about risky behaviours. Similarly, a higher practice score i.e a 1, meant malpractice, and a lower practice score i.e a 0, meant no malpractice or good practice.

- I think it would be helpful and informative if you added separate findings obtained for the knowledge, attitude, and practice sections of the survey instrument in the results section, in addition to what is currently presented.

Response: I added the extra information in the results section. See Page 4, line 36 to page 5 line 17.

- How do the results from this KAP study stand in relation to other KAP studies on human/bovine TB? Are these results similar or different?

Response: The information has been added in the discussion. E.g Page 11 line 5-17.

- Please discuss in more detail the interpretation of the results and regression analyses and the implications of these findings in context with other studies on this topic. Was the KAP survey successful in capturing knowledge gaps and data gaps among the surveyed participants in Malawi?

Response: Extra details were added in Discussion e.g on regression on Page 11, line 28-39.

- What would be some future directions as an outcome of this study?

Response: Future directions are indicated in the conclusion section.

---

## [Decision Letter · Decision Letter 1]

11 Jan 2026

Dear Dr. Ngwira,

Thank you for submitting your manuscript to PLOS ONE. After careful consideration, we feel that it has merit but does not fully meet PLOS ONE’s publication criteria as it currently stands. Therefore, we invite you to submit a revised version of the manuscript that addresses the points raised during the review process.

We look forward to receiving your revised manuscript.

Kind regards,

Rebecca Lee Smith, D.V.M., M.S., Ph.D.

Academic Editor

PLOS One

Journal Requirements:

Reviewers' comments:

Reviewer's Responses to Questions

**Comments to the Author**

Reviewer #3: (No Response)

Reviewer #4: All comments have been addressed

2. Is the manuscript technically sound, and do the data support the conclusions?

Reviewer #3: Yes

Reviewer #4: Yes

3. Has the statistical analysis been performed appropriately and rigorously?

Reviewer #3: Yes

Reviewer #4: Yes

4. Have the authors made all data underlying the findings in their manuscript fully available?

Reviewer #3: Yes

Reviewer #4: Yes

5. Is the manuscript presented in an intelligible fashion and written in standard English?

Reviewer #3: Yes

Reviewer #4: Yes

Reviewer #3: The manuscript by Ngwira et al. provides some interesting and useful results regarding the knowledge, attitudes, and practices (KAP) regarding BTB in high-burden areas of Malawi. This study demonstrates how demographics influence KAP jointly. Insights gained from this KAP assessment can inform public health officials for targeted health campaigns to reduce the spread of BTB.

Overall, the paper was well structured. There are some minor points that should be addressed.

Comment to the Editor

I found the paper interesting, and useful. I recommend acceptance and addressing my quite minor issues.

General Comments:

The manuscript by Ngwira et al. provides some interesting and useful results regarding the knowledge, attitudes, and practices (KAP) regarding BTB in high-burden areas of Malawi.

This study demonstrates how demographics influence KAP jointly. Insights gained from this KAP assessment can inform public health officials for targeted health campaigns to reduce the spread of BTB.

Overall, the paper was well structured. There are some points that should be addressed, as listed below.

Specific Comments:

Page 4, Line 2 – 4, please explain how income level was determined.

Page 4, Line 20: Add “≥” before 50%

Page 5, Line 7 – 9: If the statement implies statistical significance, please add the corresponding p-values (or confidence intervals). If no statistical test was performed, avoid using “significant” and instead use neutral phrasing such as: “A considerable proportion of participants…” or “A high proportion of participants...”

Page 5, Line 29: Mention which test was used to test significant differences in standard errors and coefficients.

Page 5, Line 39-43: If the values in table 5 are Pearson correlations among the KAP scores, the correct notation is r.

Table 1 and 2: For both the univariate and multivariate regression tables, please include p-values for each coefficient to indicate statistical significance.

Table 3: Please clarify this table. Please include p-values for each coefficient to indicate statistical significance.

S1 Appendix Question

In Background characteristics, how was income determined?

S1 Appendix Question/Statement 3 “BTB is not inherited from parents”:

The term hereditary is not appropriate because BTB is not a genetic disease; it implies a genetic predisposition, which is incorrect. Therefore, I understand that answering “Yes” would reflect correct knowledge. However, I wonder if the statement was intended to address vertical transmission instead. The translation from the African language (based on Google) reads: “Bovine TB is not acquired at birth.” If this is the intended interpretation, then “No” would be correct, since BTB can be transmitted vertically via milk or in utero.

To ensure accuracy and consistency in the manuscript, I recommend reviewing the survey translation carefully. Also, ensure consistency between the questions shown in S1 Table 1 and those in the S1 Appendix.

If the correct explanation is the former then I suggest rephrasing the results section for this question to “For Among participants, 37.87% answered ‘Yes,’ correctly recognizing that BTB is not transmitted through heredity.”

S1 Appendix, Section B

Knowledge about BTB in cattle and humans: Are questions B7, B8, B9 specific for BTB in humans or cattle? Or is both? Please clarify in page 5 line 1 and 2 “Participants also demonstrated limited knowledge on clinical signs, such as lymph node enlargement, weight loss, and low-grade fever.”

Reviewer #4: I have carefully reviewed the revised manuscript and the accompanying response to reviewers. The authors have satisfactorily addressed all the comments raised in the previous review. Revisions have been appropriately incorporated into the abstract, methods, results, and discussion, with improved clarity on the KAP framework, questionnaire development, analytical approach, and interpretation of findings. The manuscript has substantially improved in rigor, transparency, and coherence, and the responses provided are adequate and convincing. I therefore have no further substantive comments at this stage.

**Do you want your identity to be public for this peer review?** For information about this choice, including consent withdrawal, please see our Privacy Policy

Reviewer #3: No

Reviewer #4: No

---

## [Author Response · Author response to Decision Letter 2]

14 Jan 2026

Response to reviewers

Reviewer #3:

Comment to the Editor

I found the paper interesting, and useful. I recommend acceptance and addressing my quite minor issues.

General Comments:

The manuscript by Ngwira et al. provides some interesting and useful results regarding the knowledge, attitudes, and practices (KAP) regarding BTB in high-burden areas of Malawi.

This study demonstrates how demographics influence KAP jointly. Insights gained from this KAP assessment can inform public health officials for targeted health campaigns to reduce the spread of BTB.

Overall, the paper was well structured. There are some points that should be addressed, as listed below.

Specific Comments:

Page 4, Line 2 – 4, please explain how income level was determined.

Response: Participants' income was measured through their responses to the amount of money they could approximately make from different economic activities in a month. This statement has been added on page 3, line 45-46.

Page 4, Line 20: Add “≥” before 50%

Response: We thank reviewer for this suggestion. Basically, the first category is <50%, the second is 50 to < 75% and third is ≥75, which means by writing 2nd category 50 to < 75% we are already including 50, therefore ≥ before 50 may not be necessary.

Page 5, Line 7 – 9: If the statement implies statistical significance, please add the corresponding p-values (or confidence intervals). If no statistical test was performed, avoid using “significant” and instead use neutral phrasing such as: “A considerable proportion of participants…” or “A high proportion of participants...”

Response: We tried to use words like “considerable”, “substantial”, or “notable”, throughout the manuscript where there was “significant” e.g See page 5, line 4 and 5.

Page 5, Line 29: Mention which test was used to test significant differences in standard errors and coefficients.

Response: We have mentioned the test statistic that it was t-test on page 4 line 28.

Page 5, Line 39-43: If the values in table 5 are Pearson correlations among the KAP scores, the correct notation is r.

Response: We have modified by using r instead of rho, ρ. See page 5, line 32-35.

Table 1 and 2: For both the univariate and multivariate regression tables, please include p-values for each coefficient to indicate statistical significance.

Response: We thank reviewer for this suggestion. Basically, the significance of the coefficients is also determined by their 95% confidence intervals, i.e if the interval includes zero, it is considered as not significant and if the interval excludes zero, it is significant. Therefore, we still think it would suffice to have intervals only, otherwise including the p-values, we feel it will seem duplicating and overcrowding the tables. Moreover, just using the coefficient and its 95% CI is not new, see e.g Table 5 in Gebremeskel et al (2022) at https://pdfs.semanticscholar.org/a122/ca24e4f586a174847a3c78a472e418f1131a.pdf and Table 4 and 5 in Worku et al (2025) at https://www.mdpi.com/2072-6643/17/2/252.

Table 3: Please clarify this table. Please include p-values for each coefficient to indicate statistical significance.

Response: Table 3 has been moved to supplementary materials as table S4 Table according to suggestion by co-author Prof. Samuel Manda. We have clarified what is in Table 3 (now S4 Table) on page 4, line 28-30. Similarly, as in the previous response, we feel to maintain only coefficients and their 95% confidence intervals without p-values to avoid overcrowding. Significance of coefficients is determined by intervals.

S1 Appendix Question

In Background characteristics, how was income determined?

Response: Participants' income was measured through their responses to the amount of money they could approximately make from different economic activities in a month. This statement has been added on page 3, line 45-46.

S1 Appendix Question/Statement 3 “BTB is not inherited from parents”:

The term hereditary is not appropriate because BTB is not a genetic disease; it implies a genetic predisposition, which is incorrect. Therefore, I understand that answering “Yes” would reflect correct knowledge. However, I wonder if the statement was intended to address vertical transmission instead. The translation from the African language (based on Google) reads: “Bovine TB is not acquired at birth.” If this is the intended interpretation, then “No” would be correct, since BTB can be transmitted vertically via milk or in utero.

To ensure accuracy and consistency in the manuscript, I recommend reviewing the survey translation carefully. Also, ensure consistency between the questions shown in S1 Table 1 and those in the S1 Appendix.

Response: The accuracy and consistency of question items in S1 Table 1 and those in the S1 Appendix was reviewed and validated.

If the correct explanation is the former then I suggest rephrasing the results section for this question to “For Among participants, 37.87% answered ‘Yes,’ correctly recognizing that BTB is not transmitted through heredity.”

Response: The former is correct and is line with what we had initially written, nevertheless the statement has been slightly rephrased as advised by the reviewer. See page 4, line 44-46.

S1 Appendix, Section B

Knowledge about BTB in cattle and humans: Are questions B7, B8, B9 specific for BTB in humans or cattle? Or is both?

Response: We were assuming they are for both since clinical signs of BTB are indistinguishable in both cattle and humans.

See Khairullah et al (2024) at https://pubmed.ncbi.nlm.nih.gov/39055751

Please clarify in page 5 line 1 and 2 “Participants also demonstrated limited knowledge on clinical signs, such as lymph node enlargement, weight loss, and low-grade fever.”

Response: This information is based on percentages distribution in table S1 Table. In text citation of table S1 Table has been made at the beginning of the paragraph, and our understanding is that readers would be referring to this table upon reading this information. We avoid including percentages to avoid overcrowding.

---

## [Editor Report · Decision Letter 2]

15 Jan 2026

Knowledge, attitudes and practices regarding bovine tuberculosis in cattle and humans in Malawi

PONE-D-25-03110R2

Dear Dr. Ngwira,

We’re pleased to inform you that your manuscript has been judged scientifically suitable for publication and will be formally accepted for publication once it meets all outstanding technical requirements.

Kind regards,

Rebecca Lee Smith, D.V.M., M.S., Ph.D.

Academic Editor

PLOS One
---

## [Editor Report · Acceptance letter]

PONE-D-25-03110R2

PLOS One

Dear Dr. Ngwira,

I'm pleased to inform you that your manuscript has been deemed suitable for publication in PLOS One. Congratulations! Your manuscript is now being handed over to our production team.

Kind regards,

on behalf of

Dr. Rebecca Lee Smith

%CORR_ED_EDITOR_ROLE%

PLOS One